# The Protein Landscape of Mucinous Ovarian Cancer: Towards a Theranostic

**DOI:** 10.3390/cancers13225596

**Published:** 2021-11-09

**Authors:** Arkan Youssef, Mohammad B. Haskali, Kylie L. Gorringe

**Affiliations:** 1Department of Medicine, The University of Melbourne, Melbourne, VIC 3000, Australia; arkany@student.unimelb.edu.au; 2The Sir Peter MacCallum Department of Oncology, The University of Melbourne, Melbourne, VIC 3000, Australia; Mo.Haskali@petermac.org; 3Peter MacCallum Cancer Centre, Melbourne, VIC 3000, Australia

**Keywords:** mucinous ovarian cancer, theranostics, targeted therapy, personalised medicine

## Abstract

**Simple Summary:**

Mucinous ovarian cancer (MOC) is a rare type of epithelial ovarian cancer, and current treatment regimens for late stage and recurrent disease are inadequate. The ‘gold standard’ treatments are based on large clinical trials that evaluated potential therapies for all ovarian cancer, but MOC was poorly represented in these studies. As such, what works for most cases may not work for MOC. In this review, we discuss the advances in MOC treatment and explore the concept of theranostics—using therapeutic and diagnostic radionuclides against single cell surface receptors expressed highly in MOC. Additionally, we highlight the previous literature that demonstrates the overexpression of certain targets, exploring their potential to be used as theranostic targets.

**Abstract:**

MOC is a rare histotype of epithelial ovarian cancer, and current management options are inadequate for the treatment of late stage or recurrent disease. A shift towards personalised medicines in ovarian cancer is being observed, with trials targeting specific molecular pathways, however, MOC lags due to its rarity. Theranostics is a rapidly evolving category of personalised medicine, encompassing both a diagnostic and therapeutic approach by recognising targets that are expressed highly in tumour tissue in order to deliver a therapeutic payload. The present review evaluates the protein landscape of MOC in recent immunohistochemical- and proteomic-based research, aiming to identify potential candidates for theranostic application. Fourteen proteins were selected based on cell membrane localisation: HER2, EGFR, FOLR1, RAC1, GPR158, CEACAM6, MUC16, PD-L1, NHE1, CEACAM5, MUC1, ACE2, GP2, and PTPRH. Optimal proteins to target using theranostic agents must exhibit high membrane expression on cancerous tissue with low expression on healthy tissue to afford improved disease outcomes with minimal off-target effects and toxicities. We provide guidelines to consider in the selection of a theranostic target for MOC and suggest future directions in evaluating the results of this review.

## 1. Background

Mucinous ovarian carcinoma (MOC) continues to be a diagnostic and therapeutic challenge, representing a rare histological subtype of epithelial ovarian carcinoma (EOC) and contributing to 3–5% of all EOC diagnoses [1,2,3]. It is the malignant and least common form of ovarian mucinous tumours; benign cystadenomas and borderline (atypical proliferative) mucinous tumours are precursors to MOC [4]. As smoking is the only consistent risk factor for MOC, patients typically present with vague abdominal, pelvic and back pain, alongside abdominal bloating and fatigue [1]. These non-specific symptoms are thought to contribute to some patients’ presentation with large cystic masses, however, most MOC present with the tumour confined to the ovary (Stage I). Late stage disease has vastly reduced prognosis, increased risk of abdominal metastasis and recurrence [5,6,7].

Current management guidelines for MOC follow the approach of all EOC and depend on the stage of the disease at diagnosis [2,8,9]. Treatment of early stage tumour involves cytoreductive surgery (CRS) with the aim of complete resection of tumour tissue [10]. Previous studies have documented good prognosis for early stage MOC, with 5-year survival rates approaching 90% with surgery alone [2,7,11,12]. Survival between MOC and low-grade SOC (serous ovarian carcinoma, the most common histotype) is similar at stage I disease (HR, 0.87; 95% CI, 0.74–1.04) [7]. Advanced tumours receive adjuvant chemotherapy, commonly the platinum/taxane doublet, but more recently, gastrointestinal regimens such as FOLFOX, however, 5-year survival rates are poor in this population [1,13]. Five-year survival rates for stage III and IV MOC are 25.7% (95% CI, 22.9–28.7%) and 10.2% (95% CI, 8.2–12.5%), respectively, and when compared to SOC overall survival (OS) is comparatively poorer at equal stages (HR, 1.60; 95% CI, 1.48–1.73) [7]. Nevertheless, the outcomes of all late stage EOC are poor, and the standard of care is suboptimal with regard to the chemotherapy regimen and extent of cytoreductive surgery [14].

The approach to managing MOC is based on landmark randomised controlled trials that assessed potential chemotherapy treatment regimens in cohorts consisting of all EOC types, with very few MOC represented in their analyses [7,12,15,16,17]. Whilst the current approach is effective against high-grade SOC, advanced or recurrent MOC has poorer responses to standard of care [18]. It is suggested that this finding highlights the unique biological nature of MOC and its intrinsic resistance to chemotherapeutic agents [2,19].

## 2. Advances in Treatment

As it stands, there is no consensus surrounding the most appropriate treatment regimen for MOC. However, this field is evolving now, with interest in the development of different treatments in the hope of improving patient outcomes.

One such approach that has brought attention is the combination of intravenous and intraperitoneal delivery of chemotherapy following optimal CRS. Late stage EOC disseminates locally, metastasising to pelvic and peritoneal structures [20]. By delivering chemotherapy both directly to the peritoneal surface and intravenously, it is believed that a higher drug concentration can be achieved at the tumour location, thus leading to a higher residual and metastatic disease elimination [21]. In the GOG 172 Phase III trial, intravenous plus peritoneal chemotherapy demonstrated improved progression free survival (PFS) (18.3 months vs. 23.8 months; *p* = 0.05) and OS (49.7 months vs. 65.6 months; *p* = 0.03) compared to intravenous chemotherapy alone [22]. The clinical advantage of an intravenous plus intraperitoneal chemotherapy regimen is promising; however, it comes at the cost of poorer patient quality of life.

Hyperthermic intraperitoneal chemotherapy (HIPEC) is thought to improve the rates of toxicity and intolerability. By infusing heated chemotherapy into the peritoneum at select time points, improvements in disease outcome can be achieved [14,23]. A Phase III trial by Spiliotis et al. showed a 13-month improvement in mean OS (26.7 months vs. 13.4 months; *p* < 0.006) with HIPEC added to intravenous chemotherapy and CRS [24]. This three-pronged approach appears to yield favourable disease outcomes compared to standards of care, and quality of life is preserved in the long-term [25]. Although promising, these modalities have not yet been explored significantly in MOC due to the paucity of patient representation [21]. Furthermore, given the inherent differences in the biological nature of MOC, the efficacy of different chemotherapy agents, either delivered intravenously or through intraperitoneal methods, has not been determined.

Given the high level of uncertainty in the management of advanced stage and recurrent MOC, identifying promising agents is critical. Chemotherapeutic agents used in gastrointestinal tumours have for some time been investigated for their efficacy in the treatment of advanced stage or recurrent MOC based on some shared histological features [26,27]. Furthermore, molecular targeted therapies based on genetic aberrations have been of interest in the treatment of MOC, with McAlpine et al. demonstrating that *ERBB2* amplification is common in MOC, and subsequently trialling anti-HER2 therapies [28]. Common to these studies, however, has been the difficulty in recruiting participants for large scale trials and therefore drawing inconclusive results or proving limited efficacy. In the absence of larger scale combinatorial trials, the current gold-standard regimens remain in place. As such trials are unlikely to be conducted for MOC, more personalised approaches are appealing. Theranostics is one such approach that could achieve specific tumour targeting that can be approved for use based on small clinical trials feasible in a rare disease, given the ability to “see what you treat”. In addition, being able to visualise the tumour before surgery could be invaluable if a marker can be identified that is specific to carcinoma, as it is often not known before surgery whether an ovarian mass is malignant or benign.

## 3. Application of Theranostic Agents

Some developments surrounding the treatment of advanced cancers investigated the use of novel therapeutic agents that target proteins expressed by tumour tissue. One approach that does not require the inhibition of protein function is theranostics: the combination of a diagnostic test with a therapeutic intervention (Figure 1). By radiolabelling a compound of interest, commonly a peptide, small molecule, or monoclonal antibody (mAb), evaluation of drug target expression and monitoring of therapy delivery in vivo is achieved through imaging modalities such as single-photon emission computed tomography (SPECT) or positron emission tomography (PET). Candidates for therapy are first assessed using non-invasive whole-body imaging. Here, radiolabelled compounds bind to overexpressed tumour biomarkers to help identify patients who may benefit from targeted therapy. Subsequently, therapeutic beta emitting radionuclides based on the same tumour biomarker are delivered, directing radiotherapy to targets and thus tumour cells. This feature is of relevance to rare cancers, for which response can be immediately assessed in a small number of patients to provide evidence of efficacy. Peptide based radiopharmaceuticals are favoured for imaging and therapy in oncology due to their favourable pharmacokinetic profiles [29]. They offer rapid target accumulation, fast clearance from background tissue, and exhibit good tissue penetration, making them ideal candidates for theranostic applications. Where cytotoxic radiotherapy is delivered via peptides to an overexpressed target receptor, it is known as peptide receptor radionuclide therapy (PRRT) and is of clinical importance in current theranostic applications [30]. As cells express regulatory receptors that have strong affinity to a partner peptide(s), our discussion will focus on PRRT and relevant cell surface targets.

Currently, several theranostic agents (largely based on high affinity peptides) are in clinical use against other cancer types. [^177^Lu]Lu-DOTATATE/-TOC are radiopharmaceuticals that are effective in the treatment of midgut neuroendocrine tumours [31,32]. In patients who have failed first-line therapy with unlabelled somatostatin analogues, gallium-68 labelled DOTATATE/-TOC can identify patients with high SSTR expression using PET imaging. Next, therapeutic Lu-177 labelled DOTATATE/-TOC is administered, binding to the same cell-surface target receptors with similar biodistribution to the diagnostic counterpart, inducing receptor internalisation and subsequently releasing its cytotoxic payload. Compared to the standard second-line therapy of long-acting release high-dose octreotide, the NETTER-1 trial demonstrated a drastic increase in the PFS rate at 20 months, from 10.8% to 65.2% [33]. Furthermore, Lu-177 labelled urea-based peptide inhibitors of the prostate-specific membrane antigen (PSMA) have recently transformed prostate cancer treatment by delivering cytotoxic payloads systematically to all metastasis sites [34]. In prostate cancer (PCa), PSMA is overexpressed by 100–1000-fold in comparison to health prostate tissue, thereby enabling successful theranostics application [35]. Ga-68 labelled PSMA-targeting theranostics demonstrated higher diagnostic accuracy in comparison to conventional imaging techniques, leading to improved clinical management of PCa patients [36]. Moreover, Lu-177 labelled PSMA theranostics demonstrated striking responses in men with metastatic castrate resistant PCa who had progressed after treatment with conventional therapies.

A further advantage of such radioactive payloads is the ability to cause damage to nearby cancer cells (within the short range of particle emission), known as a “cross-fire” or bystander effect (Figure 1). For cells that may not be expressing the target protein, it makes resistance through downregulation of the target less likely [37]. Moreover, there is increasing interest in integrating Lu-177 radionuclide therapy (to release tumour antigens) and immunotherapy in the treatment of cancer, with the goal of long-term survival or even cure. Radionuclide therapy can lead to so-called “abscopal” effects, even in tumours with low receptor density whereby it not only shrinks the targeted tumour, but also leads to the shrinkage of untreated tumours elsewhere in the body, both locally and regionally [38,39]. Moreover, an initial reduction in tumour load can guide ongoing treatment through a process of adaptive radiotherapy; subsequent treatments for residual disease can be targeted with alpha-emitting (focal effect of action) radiopharmaceuticals [40].

Image-guided surgery (IGS) is an additional frontier in which diagnostic radiopharmaceuticals can be utilised to plan, validate, and ensure complete tumour resection, thus reducing risk of recurrence [41]. Preoperative radiotracers against a target tumour-associated antigen can be used to map and guide intraoperative surgical techniques using various imaging modalities. Then, to overcome technically or visually challenging resections and/or to confirm complete tumour removal, intraoperative visualisation using the same or similar radiotracer can assist and validate resection of the tumour, ensuring negative tumour margins and minimising surgical ambiguity. For example, in PCa, IGS with PSMA-targeting radiotracers (e.g., ^68^Ga-PSMA-11, visualised using PET) can address certain surgical challenges such as locoregional lymph node identification and confirmation of surgical margins [42]. Clinically, several established PSMA radiotracers (based commonly on γ-emission) have been evaluated and have shown effective uptake in PSMA-positive lymph nodes [42,43,44]. Moreover, CD24 is a prognostic and therapeutic tumour biomarker found in 70% of EOC cases [45]. When compared to white light and palpation, intraoperative CD-24-targeted fluorescence IGS demonstrated improved surgical outcomes in orthoptic high-grade SOC xenograft models, indicating the potential of IGS in EOC [46]. As such, diagnostic radiopharmaceuticals can support early therapeutic interventions and provide vital tools to support emerging therapeutics in the field. Furthermore, diagnostic radiopharmaceuticals against MOC can potentially be used to risk stratify patients and to guide intraoperative decisions. By identifying and validating overexpressed targets in MOC, the successes of such therapies in other tumour types can be paralleled in this disease.

Now, the question lies in identifying an appropriate tumour biomarker that can be used to treat MOC. Theranostics relies on the overexpression of target receptors on tumour cells in order to develop diagnostic and therapeutic radiopharmaceuticals. In fact, existing theranostic agents have benefited from targeting the overexpression of regulatory hormone receptors, as aforementioned. However, this process must be carefully planned. Without adequate assessment of the target’s expression in healthy, non-target tumour tissue, or the magnitude difference in expression in tumour tissue compared to non-tumour tissue, there is a risk of systemic toxicity and thus failure of drug therapy. Determining a minimum threshold of expression in tumour cells is abstract. Rather, what is necessary is to find a target with an appropriate therapeutic window where there is an increase in expression in malignant cells in comparison to background tissue as well as overexpression in an acceptable proportion of cases to justify its development. What may swing the balance towards developing a theranostic agent against a particular target is the projected utility of the drug. In diseases such as MOC where patients often relapse or fail multiple therapies and are left without effective treatment, a drug target with some off-target effects may be tolerated in the interest of preventing disease progression.

The development of high affinity theranostics with good tumour targeting is a task necessitating a multidisciplinary approach that involves medical chemists, pharmacologists, radiochemists, and biologists. Most successful theranostics currently in clinical use were developed by derivatisation of existing lead compounds with suitable characteristics. However, the development of theranostics against novel proteins that have no known lead compounds is a more complex task that requires the identification of a lead compound and its refinements towards a high affinity ligand.

## 4. Potential Theranostic Targets in MOC

We evaluated candidate targets for their potential use as theranostic targets in MOC, evaluating key characteristics such as cell surface expression and high (i.e., 100–1000×) expression in tumour epithelial cells compared to normal tissues. Ideally, candidate proteins would also have a high affinity ligand that could be exploited to develop a peptide agent. Specifically, we reviewed the literature published from 2010 onwards given the improved histopathological diagnostic techniques from around this time, whereby previously, MOC may have been misdiagnosed as mucinous metastases to the ovary [47]. The complete literature search methodology, data, and evaluation can be found in Appendix A.

We initially identified nine potential theranostic targets on the basis of reports of their expression localised to the cellular membrane: HER2, EGFR, Fra (FOLR1), RAC1, GPR158, CEACAM6, MUC16, PD-L1 (CD274), and NHE1 (SLC9A1). These candidates were evaluated through immunohistochemistry (IHC)-based analyses, yielding a semi-quantitative score that depicts their expression in MOC samples (Table 1). A discussion surrounding each of these proteins follows.

Human epidermal growth factor 2 (HER2, ERBB2) is a tyrosine kinase-type receptor that is localised to the cellular membrane, belonging to the epidermal growth factor receptor (EGFR) family [48,49]. Interestingly, whilst *ERBB1* (*HER1*, *EGFR*) is rarely amplified in MOC, *ERBB2* is amplified in roughly one quarter of all cases [1,50]. Overexpression of the HER2 oncoprotein is a predictive marker in tumour types including breast and gastric cancer where targeted therapies are in clinical use with good success [48,51]. Our search revealed numerous studies that have evaluated the expression profile of HER2 in MOC (Table 1). Notably, these studies mostly employed one of the iterations of the American Society of Clinical Oncologists and College of American Pathologists (ASCO/CAP) guidelines to quantify HER2 expression [10,52,53]. These studies each demonstrated overexpression (staining score at least 2+) of HER2 in MOC tumour samples. *ERBB2* amplification and concordance with IHC staining was assessed by Gorringe et al. using 191 MOC samples [1]. *ERBB2* was highly amplified in 26.7% of cases (51/191) and HER2 was overexpressed (staining score at least 2+) in 31.2% of cases (61/191), with strong concordance (91.8%). These data suggest a potential target for those patients stratified for HER2 expression. Indeed, in one report, monthly trastuzumab plus single agent carboplatin over six months showed a drastic tumour response in a patient with invasive MOC [28]. In another, a case of recurrent MOC was treated with trastuzumab monotherapy to good effect, normalising tumour biomarkers and improving disease status [28]. The evidence to support these benefits, although positive, is anecdotal and inconclusive, and both patients ultimately succumbed to complications of their disease. Trials are limited due to the disease’s rarity and thus the difficulty in recruiting suitable patient numbers, let alone stratifying for HER2 overexpression. A theranostic targeting HER2 could be an alternative approach using aptamers (nucleotide-based oligomers with targeting capacity) [49]. Moreover, the HER2-targeting nanobody [^177^Lu]Lu-DPTA-2Rs15d demonstrated adequate tumour imaging and reduction in murine HER2+ models [49,54]. As such, the successful chelation of both diagnostic and therapeutic moieties to a target such as a HER2-targeting nanobody or peptide may lead to advances in MOC clinical outcomes but ultimately relies on the recruitment of sufficient patients.

EGFR, another member in the same protein family as HER2, was evaluated in four studies, with variable frequencies reported (14–67%) [70,72,73,74]. Some of this variation may be due to small sample sizes, but also different scoring systems. Indeed, harmonising the scoring systems of the two largest studies to only consider at least moderate staining intensity in >10% of cells would bring them more in line (14% and 19%) [70,73]. Furthermore, it was noted that EGFR-positive MOC was associated with reduced OS and PFS (*p* = 0.02 and *p* = 0.04, respectively) [70]. Despite the relatively low rate of positive expression in MOC, a theranostic targeting EGFR is possible. Aminoflavone-loaded micelles with conjugated EGFR-targeting nanobodies demonstrated acceptable levels of cellular uptake and cytotoxicity in triple negative breast cancer cell lines, suggesting a potential nanoplatform that can be adopted for MOC [75]. Moreover, EGFR-targeting nanoemulsions carrying myrisplatin and C6-ceramide demonstrated appropriate cytotoxic effects in ovarian cancer cell-lines [76]. When trialled in platinum-resistant murine models, the nanoemulsions reduced toxicity and improved survival times when compared to conventional cisplatin treatments [77]. Although promising, these results are not stratified by histotype, and thus cannot be applied to MOC specifically.

Folate receptor alpha (Fra or FOLR1) is a member of the folate receptor family with high folic acid affinity and subsequent internalisation [78]. Its expression profile in normal tissue is restricted and localised to the cellular membrane of epithelial cells [63]. Prior literature has demonstrated elevated expression in certain carcinomas of the lung, thyroid, and breast, thus suggesting a role in tumour progression [63]. Given the significant role of folic acid in cell metabolism, and the relatively low expression in normal tissue, it can be suggested that FOLR1 plays a non-essential role in normal folate metabolism. When expressed highly in cancerous tissues, however, it may impart a proliferative advantage due to increased tumour cell folic acid requirements. The expression profile of FOLR1 was investigated in two studies with varied results, but nevertheless, demonstrated elevated expression frequencies in MOC, thus aligning with prior notions of FOLR1 overexpression in tumour cells [63]. One study with a relatively small sample size assessed FOLR1 immunoreactivity in various gynaecological cancers and found that 80% (*n* = 10) of MOC samples demonstrated positive immunoreactivity [63]. Furthermore, a second study by Kobel et al. described FOLR1 immunoreactivity in 3.1% of samples (*n* = 193) [64]. The varying rates of positivity between these two studies may largely be due to the methodological parameters that defined a positive result. In the former, samples were denoted to have positive FOLR1 expression if there was ≥5% membranous staining, whereas the latter denoted positivity as >50% membranous staining. This inconsistency in the results may warrant further FOLR1 expression analyses. To reiterate the concept of theranostics, it is not necessary for tumour cells to exhibit high rates of membranous staining (as in the latter result). Rather, theranostics relies on the expression of a target on a tumour cell so that the effect is localised to the tumour with minimal offsite effect. For cells that do not express the target, they may still be affected by the “cross-fire” effect (Figure 1). Thus, as FOLR1 may not be expressed in non-tumour tissue, it is still a valid target that warrants further investigation. To support its theranostic application, favourable clinical and dosimetric studies of folic acid-based radiotracers have shown that FOLR1 can be radiolabelled and tracked through PET with acceptable radiation dose burdens [79]. Furthermore, agents have been developed to target FOLR1 with variable results. Of note, the humanised mAb-drug conjugate mirvetixumab soravtansine showed promising results in tumours with high FOLR1 expression when assessed in Phase III trials for the treatment of platinum-resistant EOC, but not MOC specifically [78]. As such, benefit may be derived from further assessment of this target given its expression in MOC.

Ras-related C3 botulinum toxin substrate 1 (RAC1) is recognised to play a role in cellular epithelial-mesenchymal transition (EMT) through its involvement in cytoskeletal remodelling, transcriptional regulation, and cell adhesion [80]. It has previously been proposed that RAC1 overexpression is associated with melanoma progression and metastasis, and its activity linked to reduced pancreatic carcinoma cell–cell adhesion [81,82]. Moreover, colorectal tumour RAC1 expression is elevated compared to non-tumour tissue in adjacent gastrointestinal epithelia [83]. The role of RAC1 in ovarian cancer has briefly been explored, showing interactions with signalling events that mediate tumour metastasis [84]. Of particular interest to advanced stage and recurrent MOC, RAC1 overexpression has a role in chemotherapy drug resistance in other cancer types [85]. RAC1 inhibition reduces EGFR-TKI drug resistance in gefitinib-resistant non-small-cell lung carcinoma [86]. In addition, TIAM1-overexpressed multidrug-resistant lymphoma cell lines targeted with dual TIAM1-RAC1 and NOTCH pathway inhibitors improved sensitivity to adriamycin [87]. Clinically, targeting the TIAM1-RAC1 pathway in fludarabine-resistant chronic lymphocytic leukaemia can improve fludarabine sensitivity [88]. Targeting RAC1 overexpression in MOC may yield favourable results in advanced disease and is supported by findings that may suggest treatment sensitisation in cisplatin-resistant gastric adenocarcinoma cell lines [89]. Leng et al. demonstrated a significantly increased expression frequency of RAC1 in a population of MOC when compared to normal ovarian samples (55%, *n* = 40 vs. 4.2%, *n* = 24; *p* < 0.05), supporting a theory that high expression levels of RAC1 may correlate with advanced disease stage and poorer prognosis in MOC [65]. Given the evidence that supports RAC1 activity inhibition in various tumour types and its elevated expression in MOC, there may be potential in not only targeting this receptor with theranostic agents, but also in exploring agents to inhibit it in treatment-resistant and advanced MOC with a view of improving sensitivity to current chemotherapy agents.

GPR158 is a cell-surface signalling molecule belonging to the G Protein coupled receptor (GPCR) superfamily [66]. GPCRs are a widely studied therapeutic target, as their activation has been related to broad physiological and disease processes [90]. Specifically, the expression and upregulation of GPR158 has been linked to tumour progression and unfavourable survival in prostate cancer and gliomas, but the literature is otherwise limited with regard to targeted GPR158 therapy [66,90]. Its association with MOC was explored by Engqvist et al., wherein 59% (*n* = 29) of tumour samples demonstrated positive expression [66]. Although this finding may suggest a promising therapeutic target, a semiquantitative score that combined staining proportion and intensity was positive if the score was greater than 0 (range 0–300), suggesting a lack of stratification of tumour samples based on the score. Most MOC cases showed only low intensity expression. Furthermore, there are no data on exploring the protein’s expression in normal, non-tumour tissue, apart from high protein expression observed in the brain, which would have to be considered in the design of a theranostic [91]. Given the paucity of literature exploring GPR158, it would require further evaluation in the form of expression quantification in both MOC and non-tumour tissue in order to further evaluate its potential as a theranostic.

Nonspecific cross-reacting antigen (NCA-90, CEACAM6), a member of the carcinoembryonic antigen gene family, is expressed on granulocytes and epithelial tissue [67]. It is a complex cell-surface glycoprotein macromolecule that is involved in cell adhesion and tumour progression, implicated particularly in the context of cytokine-activated neutrophil-endothelial cell adhesion [67]. In the absence of adhesive interactions with the extracellular matrix, normal cells undergo anoikis. CEACAM6 inhibits this process, and thus higher expression has been correlated to loss of tumour cell differentiation—a process typical and characteristic of tumorigenesis and metastasis [92,93]. Antibodies directed against its N-domain alter intercellular interactions, thus suppressing the malignant potential of tumours [67,94]. Of note, CEAMCAM6-targeting albumin-based nanoparticles have demonstrated efficacy in targeting CEACAM6 in anoikis-resistant tumour cells [95]. Moreover, these nanoparticles have been effective in reducing the metastatic capacity of anoikis-resistant lung carcinomas in mice [67,95]. CEACAM6 immunoreactivity was compared between eight MOC cases (88% positive) and ten cases of normal ovarian tissue (0% positive) [67]. Given the in vivo, albeit in mouse models, efficacy of a nanoparticle against CEACAM6, further evaluation of this target’s expression in MOC is warranted.

MUC16, or CA125, is a highly evaluated tumour biomarker that is elevated in ovarian cancer and is linked with disease progression. It is used as a clinical serum biomarker for diagnosis and for tracking, following disease burden after surgery and systemic treatment. It is a highly glycosylated membrane-bound mucin that is expressed by the epithelial lining of various anatomical structures, acting as a barrier and maintains surface mucosa [68,96]. Its expression profile was investigated by Vitiazeva et al., whereby four out of the six cases of MOC (67%) demonstrated moderate MUC16 expression. The limitation of this biomarker, however, is its low sensitivity and specificity to MOC as seen in other gynaecological and non-malignant diseases and thus limited role in theranostics [96].

Programmed cell death protein like 1 (PD-L1) is an inhibitory ligand found commonly in the tumour microenvironment, playing a role in inhibiting immune regulatory mechanisms. When bound to programmed cell death 1 (PD-1) on activated CD4 and CD8 T cells, it suppresses subsequent immune interactions, thus inducing a tumour naïve response [69]. In some tumours such as gastric cancer and hepatocellular carcinoma, its overexpression is correlated with poorer prognostic outcomes, but interestingly, positive clinical outcomes are observed in breast cancer [97]. In ovarian cancer, the prognostic value of PD-1 is not yet fully understood, but has been associated with improved OS in high-grade SOC [97]. As such, it is an important clinical target and biomarker for some cancer types being treated with immune checkpoint inhibitors. Its expression, however, is not restricted to only tumour cells, having documented roles in many cell types [69,98]. PD-L1 may be an inappropriate target for theranostics, although this study raises the question of whether MOC could be targeted by immune checkpoint inhibitors. However, the effectiveness of PD-1/PD-L1 blockade in ovarian cancer has seen limited success and is highly variable between patients and histotypes—the cause of this variability is poorly understood, but is perhaps linked to the degree of genomic instability [99].

Sodium hydrogen exchanger 1 (NHE1, SLC9A1, solute carrier family 9 member A1) is a ubiquitous cell-surface multi-pass membrane protein and a major pH regulator with roles also in signal transduction in the kidneys and intestines [83]. It belongs to the sodium hydrogen exchanger (NHE) family [100]. Intracellular pH derangements are associated with cellular transformations characteristic of tumours for which NHE1 may play a part in, with downregulation of NHE1 imparting tumour suppressive effects in gastric cancer and gliomas [100,101,102]. Previously, NHE1 overexpression was shown to be present in treatment resistant breast cancer, and inhibition of NHE1 with cariporide induced sensitivity to doxorubicin, thus suggesting a therapeutic target [101]. Moreover, NHE1 was found to be expressed in 79.7% (*n* = 129) of EOC compared with only 26.7% (*n* = 15; *p* < 0.01) in normal ovarian tissue, with no significant difference in expression between EOC subtypes [103]. NHE1 expression was correlated with tumour progression and advanced EOC, suggesting a potential predictive biomarker [103]. Tang et al. evaluated autoantibodies that target integrins and Wnt signalling pathway antigens in MOC plasma samples [71]. Subsequently, these antigen targets were analysed through IHC to evaluate their expression profiles in tissue samples. Of the panel assessed, all but one of the nine antigens displayed significantly elevated expression in MOC tissue samples (*n* = 15) when compared to normal ovarian tissue. NHE1 was the only antigen localised to the cell membrane. The difficulty with this finding, however, is that staining scores were omitted. As such, the intensity and proportion of NHE1 staining in the samples cannot be appreciated quantitatively. Nonetheless, NHE1 is a cell-surface protein that is involved in cell pH regulation and in MOC, this pathway may be deregulated with significantly increased expression.

IHC is a technique commonly used to explore target protein presence and expression profiles in a histological sample. By nature, this approach requires a predetermined target and is inherently biased. In fact, data on expression are often described using semi-quantitative measurements and thus are not quantitatively as accurate as other molecular analyses [37]. A further literature search was performed to identify records that utilised high-throughput proteomic analyses in order to objectively reveal potential targets that are expressed aberrantly in comparison to a reference proteome as well as to accurately quantify their expression. The targets that we will discuss now are: CEACAM5, CEACAM6, MUC1, ACE2, GP2, and PTPRH (Table 2).

Each of these targets were quantified by Tian and colleagues [104]. Here, targets were assessed via glycoproteomic analysis and quantified through spectral counting. A liquid chromatography mass spectrometry process was used to determine the presence of peptides within samples, with quantification of protein abundance by each peptide’s spectral count. Spectral counting is a label-free method, as opposed to the gold standard label-dependent method, in which the abundance of identified peptides are measured without the limitation of a fixed number of channels from the labelling agent [105]. A ratio of spectral counts between MOC (*n* = 3) and normal ovary (*n* = 3) was determined for these peptides, where an arbitrary score of 100 indicated the presence of a peptide only in MOC and not in normal tissue. Six glycopeptides scored 100 and were identified as having altered expression in MOC compared to normal ovarian tissue. Despite this, some targets are expressed ubiquitously or in seemingly substantial levels in normal tissue as supported by The Universal Protein Resource [83]. Ubiquitously low expression of a target is mandatory for successful theranostic application to avoid off-target toxicity and to obtain high tumour to target ratio.

Carcinoembryonic antigen (CEA, CEACAM5) and CEACAM6, as aforementioned, share antigenic similarities and function similarly in their ability to modulate the routine cell process of anoikis [67,92]. The proteomic analysis was validated through western blotting, where their expression was confirmed to be elevated in MOC [104]. Both proteins have been observed to be expressed in multiple cancer types, and despite some expression in normal gastrointestinal tract tissues, the levels in tumours were elevated 2–4 fold, depending on the tissue type [93]. Indeed, there are several clinical trials currently in progress of therapeutic antibodies (not as theranostics) to both proteins [106,107]. The mAb NEO-201 with selective CEACAM5 and CEACAM6 targeting capacity and anti-tumour activity showed strong immunoreactivity in a cohort of MOC samples and other tumour types with minimal cross-reactivity to surrounding normal tissue [108]. Additionally, NEO-201 was previously shown to exhibit antibody-dependent cellular cytotoxicity in pancreatic and ovarian tumour models with tolerable safety profiles, thus progressing to Phase I human clinical trials [108,109]. In castrate-resistant neuroendocrine prostate cancer, antibody-drug conjugates targeting the cell surface antigen CEACAM5 have shown strong evidence in clinical trials [110].

Tumour-associated Mucin-1 (MUC1) is overexpressed on the cell surface of more than 90% of EOC and facilitates cancer progression and metastasis [111]. It is structurally different to normal MUC1, most notably and in the interest of targeted therapy by its altered glycan chains, which revealed a novel epitope in the protein core. In the normal cell, this would otherwise be masked by glycosylation [111,112]. Higher expression of tumour-associated MUC1 has been linked to micro-metastasis formation, where it functions as an anti-adhesion molecule, allowing for the migration of tumour cells [113]. As such, tumour-associated MUC1 may be targeted due to its inherent structural changes found only in tumour tissues. To support this finding, several IHC-based studies evaluated MUC1 expression in MOC and demonstrated elevated expression frequencies [114,115,116]. However, only one study by Van Elssen differentially explored the expression of tumour-associated MUC1; other studies did not comment on the structural differences to MUC1 and thus may not be truly representative of tumour-associated MUC1 in MOC. Nevertheless, there are preclinical and clinical trials assessing anti-tumour-associated MUC1 agents in ovarian carcinomas [111]. Notably, the conjugated murine mAb C595 (targeting the core epitope) has shown success in radiolabelling and identifying tumour-associated MUC1-overexpressed EOC [111]. Furthermore, the C595 mAb combined with docetaxel showed dose-dependent efficacy in inducing tumour sensitivity and inhibition of tumour progression in EOC cell lines in vitro [117].

Pancreatic secretory granule membrane major glycoprotein GP2 (GP2) is produced and released from pancreatic zymogen granules, and is involved in the innate immune response [83]. Moreover, receptor-type tyrosine-protein phosphatase N2 (PTPRN2) is a transmembrane peptide that appears to have a distinct role in endocrine and neuronal vesicle-mediated secretory processes [83]. Its downregulation has previously been associated with defects in cell invasion in highly metastatic breast cancer [118]. Whilst the expression of both peptides appears elevated in MOC tissue samples, there is a paucity of literature that explores their role in tumorigenesis or validates their expression in normal tissue or MOC. In addition, there is no identifiable literature that explores their therapeutic potential, thus warranting ongoing evaluation.

Angiotensin-converting enzyme 2 (ACE2) has established roles in the renin-angiotensin hormone system as a carboxypeptidase responsible for the conversion of angiotensin I and angiotensin II to angiotensin 1–9 and 1–7, respectively [83]. Further assessment of its level of expression in MOC is required given its ubiquitous expression in non-malignant tissue. This expression feature may reduce its potential as a theranostic target.

## 5. Conclusions

EOC represents a highly heterogenous group of gynaecological cancers with differences in their clinicopathological features. Treatment, therefore, must be tailored to the specific subtype. The management of MOC is rapidly evolving with focus now on personalised medicines including molecular therapies, IGS, and theranostics. Theranostics is a new area that has not been explored for this disease. In this review, we highlighted several targets that show promising and potential applicability within the realm of theranostics. However, the findings suggest mere associations between expression frequencies and clinical stage of disease, and positive expression profiles are frequently not scored using consistent scoring systems. As such, we propose that future research employs standardised scoring systems, when applicable, for the scoring of protein expression profiles. Furthermore, our search strategies may not have been all-encompassing. To combat this, multiple search strategies were employed in the interest of ensuring the robustness and completeness of this review, however, we cannot be certain that we did not miss a potential target. Finally, the next step forward builds on this literature and lies in evaluating the expression incidence and levels of these targets in normal tissue. In all of these studies, the primary aim was to elucidate protein expression profiles in MOC only. In order to verify theranostic applicability, characterising the location, specificity, and magnitude of protein expression in non-MOC samples is crucial in predicting non-therapeutic side effects. In addition, the identification and optimisation of a targeting molecule such as a high affinity binding peptide and selection of a radionuclide tracer are key steps in taking a target towards the clinic.

## Figures and Tables

**Figure 1 cancers-13-05596-f001:**
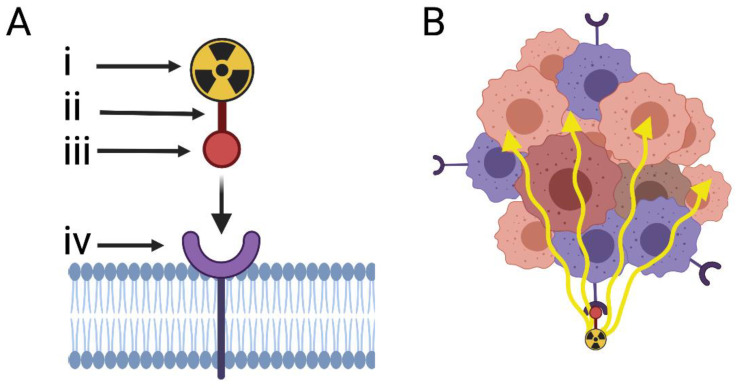
(**A**) Diagrammatic illustration of the two-step utility of a theranostic agent against a targeted cell-surface receptor. The theranostic agent is composed of a single targeting peptide (iii) that has affinity to the target receptor (iv) and is linked via a binding molecule (ii) to the diagnostic or therapeutic radionuclide (i). Diagnostic radionuclides are often positron emitting (e.g., Ga-68) and therapeutic radionuclides are often beta (e.g., Lu-177) or alpha emitting (e.g., Ac-225). For either radionuclide types, the targeting peptide remains the same. (**B**) The bystander effect. A therapeutic agent (alpha-/beta-emitting radionuclide conjugated to a receptor-seeking peptide) first binds to a cell surface tumour receptor, then creates a cytotoxic environment in which surrounding tumour cells that do not express the receptor (orange cells) are affected. Created with BioRender.com.

**Table 1 cancers-13-05596-t001:** Immunohistochemical expression profiles of selected targets in MOC.

Protein of Interest	Cell Type	Subcellular Localisation ^1^	Cases of MOC	Control Tissue Expression	Expression in MOC
**HER2**					
Anglesio [55]			154	-	29/154 (18.8%) ^2^
Bassiouny [56]			36	-	6/35 (17.1%) ^3^
Chapel [57]			6	-	2/6 (33.3%) ^4^
Mohammed [48]			20	0/30 (0%)	11/20 (55.0%) ^3^
Chen [58]	Epithelial	CM, E, C, N	49	-	11/49 (22.4%) ^3^
Chao [59]			49	-	9/49 (18.4%) ^3^
Missaoui [60]			14	-	2/14 (14.3%) ^2^
Kim [61]			46	-	14/46 (37.84%) ^2^
Yan [62]			17	-	5/17 (29.4%) ^5^
**Fra/FOLR1**					
O’Shannessy [63]	Epithelial	S, CM, E	10	-	8/10 (80%) ^6^
Kobel [64]			193	-	6/193 (3.1%) ^7^
**RAC1**					
Leng [65]	Epithelial	CM	40	1/24 (4.2%) ^8^	22/40 (55%) ^8^; *p* < 0.05
**GPR158**					
Engqvist [66]	Epithelial	CM	29	-	17/29 (59%) ^9^
**CEACAM6**					
Lee [67]	Epithelial	CM	8	0/10 (100%) ^10^	7/8 (88%) ^10^
**MUC16**					
Vitiazeva [68]	Epithelial	CM	7	-	4/6 (66%) ^11^
**PD-L1**					
Webb [69]	Various	CM	30	-	8/30 (27%) ^12^
Hada [70]			49	-	12/49 (24.5%) ^13^
**NHE1/SLC9A1**					
Tang [71]	Epithelial	CM, ER	15	-	Elevated ^14^ (*p* = 0.002)
**EGFR**					
Hada [70]			49	-	7/49 (14%) ^13^
Cirstea [72]			7	-	3/7 (43%) ^15^
Alshenawy [73]	Epithelial	S, CM	21	-	10/21 (47.6%) ^16^
Tanaka [74]			3	-	2/3 (67%) ^17^

^1^ Abbreviations: CM, cell membrane; S, secreted; E, endosome; ER, endoplasmic reticulum; N, nuclear. ^2^ Scored according to guidelines from the American Society of Clinical Oncologists and College of American Pathologists (ASCO/CAP) 2007. ^3^ Scored according to guidelines from the American Society of Clinical Oncologists and College of American Pathologists (ASCO/CAP) 2013. ^4^ Scored according to guidelines from the American Society of Clinical Oncologists and College of American Pathologists (ASCO/CAP) 2018. ^5^ Samples were denoted positive if moderate-strong membranous staining was exhibited in >10% of cells per sample. ^6^ Proportion of positively staining samples, where membranous staining in ≥5% of tumour cells in a sample at any staining intensity >0 (scores ranged between 0 and 3+, based on level of magnification to confirm staining) was considered positive. ^7^ Samples were denoted positive if there was strong membranous staining (>50% of tumour cells). ^8^ Proportion of samples in which there was high expression. A semiquantitative score, calculated out of nine and comprising the product of staining intensity and percentage of positive tumour cells in a sample, was used. For each sample, the score was used to determine if expression was absent, low or high. High expression represented a score ≥5. Statistically significant in comparison to normal stromal tissue. ^9^ Proportion of samples in which staining was considered positive. A semiquantitative immunoreactive score (H-score) was used, and samples with scores >0 (range 0 to 300) were considered positive for target protein expression. The H-score was based on the proportion and intensity of positively stained tumour cells. ^10^ Proportion of samples in which staining was considered positive. Staining that was observed within the luminal cell border or cytoplasm in ≥10% of cells within the tissue section was considered positive. ^11^ Proportion of samples in which staining was considered positive. A semiquantitative score, out of 3, was determined equally based on intensity and proportion of staining in a 0.5 cm^2^ tumour sample. The proportion of tumour cells was scored between 0 and 3, whereby a score of 1 or more represented at least 1/3 cells stained. The intensity of sample staining was scored between 0 and 3, where 0 was none and 3 was strong. Average score for proportion of cells stained in positively staining cells was 2.25. Average intensity of staining in positive samples was 2.75. ^12^ Proportion of samples in which staining was considered positive, using a threshold of ≥1 positive cell for each tissue microarray core. ^13^ Proportion of samples in which staining was considered positive. Samples were positive if there was >10% staining of cells per sample with staining intensity either 2 or 3. ^14^ Denotes statistically significant expression in tissue samples of mucinous ovarian cancer compared to normal ovarian tissue. Samples were assessed semi quantitatively based on staining proportion and intensity, where the maximum score was 9. The staining and thus difference between tumour and normal tissue samples was compared for statistical significance (*p* < 0.05). Specific scores were not provided, but a clear scoring methodology and illustrations of staining were provided to support the findings. ^15^ Proportion of samples with >5% staining of any intensity. ^16^ Proportion of samples with >25% membranous staining at least weak intensity. ^17^ Proportion of samples with >70% cytoplasmic, nuclear, or membranous staining.

**Table 2 cancers-13-05596-t002:** Proteomic expression profiles of the selected proteins, with comparison to a control proteome.

Protein(s) of Interest	Subcellular Localisation ^1^	Expression in MOC ^2^
Tian, Y (2011) et al. [104] **CEA cell adhesion molecule 5 (CEACAM5)** **CEA cell adhesion molecule 6 (CEACAM6)** **Mucin-1 (MUC1)** **Angiotensin-converting enzyme 2 (ACE2)** **Pancreatic secretory granule membrane major glycoprotein GP2 (GP2)** **Receptor-type tyrosine-protein phosphatase N2 (PTPRN2)**	CM, CS CM, CS CM S, CM, C CM, S CM, C	100.00 100.00 100.00 100.00 100.00 100.00

^1^ Abbreviations: CM, cell membrane; CS, cell surface; S, secreted; C, cytoplasm. ^2^ Targets identified through MS/MS spectra were searched in UniProt. A ratio of spectral counts comparing expression in normal tissue (Human International Protein Index (UniProt), and MOC. An arbitrary count of 100.00 was assigned to proteins that demonstrated expression only in MOC.

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
