# Peer review of "The Protein Landscape of Mucinous Ovarian Cancer: Towards a Theranostic"

_cancers, 2021, doi:10.3390/cancers13225596_

Round 1
Reviewer 1 Report
The manuscript has improved much through the revision process.
Major:
The use of theranostics for image-guide surgery needs a better focus throughout the manuscript as it is still hardly mentioned.
Minor:
As the no consensus regarding the best treatment regimen exists the statement p2, l 50-51 should be modified
Author Response
Comment:
The manuscript has improved much through the revision process.
Major:
The use of theranostics for image-guide surgery needs a better focus throughout the manuscript as it is still hardly mentioned.
Response:
As well as including IGS in the conclusion, the following paragraph has been added on page 4:
“Image-guided surgery (IGS) is an additional frontier in which diagnostic radiopharmaceuticals can be utilized to plan, validate and ensure complete tumour resection thus reducing risk of recurrence [40]. Preoperative radiotracers against a target tumour-associated antigen can be used to map and guide intraoperative surgical techniques using various imaging modalities. Then, to overcome technically or visually challenging resections and/or to confirm complete tumour removal, intraoperative visualization using the same or similar radiotracer can assist and validate resection of the tumour, ensuring negative tumour margins and minimising surgical ambiguity. For example in PCa, IGS with PSMA-targeting radiotracers (e.g. 68Ga-PSMA-11, visualized using PET) can address certain surgical challenges such as locoregional lymph node identification and confirmation of surgical margins [41]. Clinically, several established PSMA radiotracers (based commonly on g-emission) have been evaluated and have shown effective uptake in PSMA-positive lymph nodes [41-43]. Moreover, CD24 is a prognostic and therapeutic tumour biomarker found in 70% of EOC cases [44]. When compared to white light and palpation, intraoperative CD-24-targeted fluorescence IGS demonstrated improved surgical outcomes in orthoptic high-grade SOC xenograft models, indicating the potential of IGS in EOC [45]. As such, diagnostic radiopharmaceuticals can support early therapeutic interventions and provide vital tools to support emerging therapeutics in the field. Furthermore, diagnostic radiopharmaceuticals against MOC can potentially be used to risk stratify patients and to guide intraoperative decisions. By identifying and validating overexpressed targets in MOC, the successes of such therapies in other tumour types can be paralleled in this disease”
Minor:
As the no consensus regarding the best treatment regimen exists the statement p2, l 50-51 should be modified
Modified to:
“Advanced tumours receive adjuvant chemotherapy, commonly platinum/taxane doublet and more recently gastrointestinal regimens such as FOLFOX,”
This manuscript is a resubmission of an earlier submission. The following is a list of the peer review reports and author responses from that submission.
Round 1
Reviewer 1 Report
This is an interesting review of the literature in order to identify potential theronostic targets for mucinous ovarian cancer. The authors identified from their review of the literature, 13 proteins that are cell surface localized and discusses each of these as potential therapeutic targets by theronistics.
Some comments:
1) The methods should have a clear statement of the inclusion and exclusion criteria for the triaging their review studies from the 3 online databases. This is included in the abstract, and is only referred to in the methods.
2) There needs to be a more detailed section on the management of MOC, with reference to hazard ratios for death, response rates to chemotherapy, overall survival and progression free survival. This will enable to reader of the review to understand the current shortfalls in management strategies, and hence the need for more precise, tumor specific therapies, such as a theronostic approach.
3) I am surprised that the authors have not selected epidermal growth factor receptor as a potential target. According to some published reports which should have passed their selection criteria, EGFR is amplified in MOC.
Similarly, the authors have not commented on the potential theronostic targeting of HER2 given that amplification of HER2 has been reported by their own research group. There is only a minor mention in the introduction on trialling anti-HER2 therapies.
There is a body of literature describing theranostic targeting of both EGFR and HER2. This review would be strengthened by including a review of these.
Reviewer 2 Report
The authors present in this review manuscript a systematic overview of possible molecular targets that can be used as theranostics for mucinous ovarian cancer. The subject for the review is timely but I have identified some inaccuracies that must be corrected before the manuscript can be recommended for publication. The field of theranostics is so much broader than what the authors cover in their text throughout the manuscript.
Abstract
I will suggest that the document is written up according to “the instructions to authors”:
First a simple summary is presented. This is followed by an abstract – no subheadings are used here.
General comment:
The field of theranostics is much broader than what the authors cover in their text throughout the manuscript. The parts focusing on theranostics must be updated. The role theranostics might play for image-guided surgery is hardly mentioned.
Introduction
- See general comment.
- The role surgery play at advanced stage disease should be highlighted. Maybe also some words about HIPEC.
- What do you believed can be achieved with the use of theranostics ? This must be added.
- A paragraph describing the useful characteristics theranostic biomarkers in general should also be added.
Results
A detailed overview of which cell types that express the marker is missing. In the introduction and discussion this information should be highlighted as relevant for biomarker selection.
Discussion
- See general comment.
- The clinical relevance of the different biomarkers should be high-lighted - not only the potential but also clinical experience.
- The selection of tracers is a subject by itself and beyond the scope of this paper, but should be mentioned.
Reviewer 3 Report
This is an excellent review describing very important information on upregulated biomarker expression in mucinous ovarian cancer. The review, in my opinion, is significant, not only because it brings together information on the relative expression of the various receptors that are upregulated in this disease, but provides invaluable information to other scientists also working in the area on fine-tuning potential targets for radiopharmaceutical (theranostic) development.
The review is well written and the basis for the upregulated receptor expression for the literature is sound considering the difficulty in patient recruitment and the rarity of this cancer.
Author Response
Many thanks to the reviewer for their positive comments